# Detecting Unique Analyte-Specific Radio Frequency Spectral Responses in Liquid Solutions—Implications for Non-Invasive Physiologic Monitoring

**DOI:** 10.3390/s23104817

**Published:** 2023-05-17

**Authors:** Dominic Klyve, James H. Anderson, George Lorentz, Virend K. Somers

**Affiliations:** 1Department of Mathematics, Central Washington University, Ellensburg, WA 98926, USA; 2Know Labs Inc., Seattle, WA 98101, USA; andy@knowlabs.co; 3Mayo Clinic, Rochester, MN 55902, USA; lorentz.george@mayo.edu (G.L.); somers.virend@mayo.edu (V.K.S.)

**Keywords:** biosensors, clinical applications, point-of-care, radio frequency sensors, non-invasive blood glucose monitoring, microwave sensors

## Abstract

With rising healthcare costs and the rapid increase in remote physiologic monitoring and care delivery, there is an increasing need for economical, accurate, and non-invasive continuous measures of blood analytes. Based on radio frequency identification (RFID), a novel electromagnetic technology (the Bio-RFID sensor) was developed to non-invasively penetrate inanimate surfaces, capture data from individual radio frequencies, and convert those data into physiologically meaningful information and insights. Here, we describe groundbreaking proof-of-principle studies using Bio-RFID to accurately measure various concentrations of analytes in deionized water. In particular, we tested the hypothesis that the Bio-RFID sensor is able to precisely and non-invasively measure and identify a variety of analytes in vitro. For this assessment, varying solutions of (1) water in isopropyl alcohol; (2) salt in water, and (3) commercial bleach in water were tested, using a randomized double-blind trial design, as proxies for biochemical solutions in general. The Bio-RFID technology was able to detect concentrations of 2000 parts per million (ppm), with evidence suggesting the ability to detect considerably smaller concentration differences.

## 1. Introduction

Non-invasive testing for chemicals and disease detection in humans has existed for over 3000 years. Ayurvedic physicians in the sixth/fifth century BCE first described a sweet taste in urine, naming the condition madhumeha (honey urine) [1], which was later identified as glucose excreted in the urine of individuals with diabetes mellitus.

While non-invasive detection of chemicals has hundreds of possible useful physiological applications, quantifying glucose and other analytes in the blood remains a high priority. This testing has taken a variety of forms over time. In the 20th century, non-invasive testing left the realm of the human sensorium, and became rooted in technology, with electromagnetic sensing technology using microwaves proving a particularly useful method for molecular detection.

The emergence of microwave sensors can be traced back to the pioneering work of Arthur von Hippel (co-developer of radar during the Second World War) in the 1950s, when the physicist developed improved techniques to measure permittivity and to investigate the relationship between permittivity and the physical properties of materials and mixtures [2]. Initially, microwave sensors were constrained in their utility due to the scarcity, high cost, and cumbersome nature of components, as well as the limited processing capabilities for signals. As a result, their practical applications were limited. However, with the increasing availability of solid-state components in the 1970s, the situation changed. In the early 1980s, the integration of microprocessors into measurement devices made it possible to create simple yet sophisticated microwave sensors that were affordable. Consequently, the range of applications and commercial products for microwave sensors has since expanded rapidly [3].

Researchers have demonstrated the ability to use microwaves to identify other molecules. Various alcohols have attracted much of this attention. For example, RF methods have proven useful in detecting methanol concentrations in ethanol [4], methanol concentrations in water [5], and the microwave rotational spectra of other alcohols [6,7,8,9,10]. Similarly, RF methods have been used to determine the moisture content of stone materials [11] and to analyze and detect silver materials in an aqueous solution [12]. Clearly, RF spectroscopy—especially within the microwave band—has shown a variety of industrial and medical applications. In this work, we have examined the ability of a novel type of sensor to use RF methods for non-invasive quantification of several different chemical solutes in a liquid analyte.

This study is motivated by the ultimate goal of measuring blood glucose levels (BGL) in vivo. It was performed using a novel electromagnetic sensor technology (the Bio-RFID sensor) to non-invasively penetrate surfaces, capture data from individual radio frequencies, and convert those data into physiologically meaningful information and insights. However, measuring glucose comes with many challenges. Among others, blood glucose is subject to degradation as soon as it is removed from a biological system. In this work, we report on the first formal experiments conducted using the Bio-RFID sensor, and describe a series of experiments in which we quantify solutes that do not degrade, in order to establish proof of principle in vitro, with the hope of moving on to biological compounds.

## 2. Materials and Methods

### 2.1. Study Design

An observational study design was used in which data were collected and analyzed during the period 3–5 March at the St. Mary’s campus of Mayo Clinic in Rochester, MN. Data were collected from a series of liquid analytes using the sensor described below. The study involved a series of five experiments, each of which demonstrated the ability of the sensor to non-invasively quantify concentrations of a solute in liquid by scanning a series of solutions once, and then performing blinded scans of the same solutions.

### 2.2. The Sensor

The sensor employed in this study was the patented Know Labs Bio-RFID device, a sensor that generates RF signals and measures received power through an antenna array. Engineering details of the sensor are given in [13]; here, we give a high-level summary of its functionality. The sensor can generate RF signals that range from just over 100 MHz to just over 4000 MHz, and has a transmit amplifier to boost the signal. The RF signal is routed through a switch matrix that allows it to be sent to any one of the four supported antenna elements or through an onboard fixed-attenuation path called the calibration path that allows the system to test itself and provides a known benchmark. (Although neither was used in this study, the sensor is also equipped with an onboard inertial measurement unit (IMU) to detect motion and a thermometer to compensate for any temperature-related signal drifts). A microcontroller accumulates samples during a dwell, which is defined as the period of time that the signal generator is active at a specific frequency, and then averages those samples to provide a single power measurement value per dwell over the device’s USB connector.

### 2.3. Physical Setup

The in vitro Bio-RFID sensor consists of an acrylic cuvette in the shape of a 4-inch × 4-inch × 8-inch rectangular prism mounted on the Bio-RFID sensor. This apparatus is in turn mounted on a balance that allows the determination of the mass of the analyte being scanned to within 0.1 g. The sensor scans through several thousand radio frequencies and sends the data to custom software on a laptop computer.

The software displays a spectral scan of the analyte (its Bio-RFID signature) and computes a similarity metric for each of the solutions in the training data.

### 2.4. Study Protocol

With the goal of demonstrating the sensor’s ability to quantify a range of analytes non-invasively, the following study design was established and employed in a series of experiments. Each experiment involved choosing a solvent (deionized water or isopropyl alcohol) and a solute. Solutions were then prepared with differing concentrations of the solute. Internal experiments demonstrated that temperature could have a measurement impact on the shape of the Bio-RFID curves. We thus attempted to reduce this impact by maintaining a temperature as close to 25 °C as possible. In experiments 1–4, a training set was created by scanning each of the solutions, followed by a blinded test in which the team identified the analyte based on the signature. Experiment 5 could not be performed blind, but yielded results comparable to the other tests.

Specifically, we employed the following:Procedure for Blind Identification
1.Each of the solutions was prepared and then kept in a bottle with a labeled lid.2.Each of the solutions was scanned by the Know Labs team, with full knowledge of which analyte was being scanned, and the Bio-RFID signature was captured to create the training dataset.3.All solutions were removed from the testing space by independent laboratory personnel.4.Laboratory personnel randomized the solutions, removed the identifying covers, and brought solutions, one at a time, to the Know Labs team.5.The Know Labs team transferred the solution to their cuvette, scanned it with the Bio-RFID sensor, and attempted to identify the solution.

A total of five experiments were conducted: three of various concentrations of deionized water in isopropyl alcohol, one of various concentrations of commercial NaCl in deionized water, and one of various concentrations of commercial bleach in deionized water.

#### 2.4.1. Experiment 1: Deionized Water in Isopropyl Alcohol at 10,000 ppm

Experiment 1 was designed to demonstrate the technology’s ability to measure differences of concentrations of water in isopropyl alcohol (for two solutions with equal masses) at a level of 10,000 ppm = 1%.

##### Solutions Tested

Six liquid solutions of commercially available “99% isopropyl alcohol” (from the same lot) and deionized water, containing 0%, 1%, 2%, 3%, 4%, and 5% deionized water by volume in isopropyl alcohol were tested. In the following, these mixtures are referred to simply by their concentration of deionized water.

##### Preparation of Solutions

Six individual containers were used, each with a volume of 3940 mL. To make the 0% solution, the first container was filled with 99% isopropyl alcohol. The 1% solution was prepared by filling the second container with 3940 mL of 99% isopropyl alcohol, after which 39.4 mL was removed and replaced with 39.4 mL of deionized water of the same temperature. Other concentrations were prepared similarly, removing 78.8 mL, 118.2 mL, 157.6 mL, and 197.0 mL, respectively, of isopropyl alcohol and replacing it with deionized water.

##### Identification of Blinded Samples

The team then followed the “Procedure for Blind Identification” described above using 1000.0 g of each of the prepared solutions.

#### 2.4.2. Experiment 2: Deionized Water in Isopropyl Alcohol at 2000 ppm

Experiment 2 was similar to Experiment 1 in that it demonstrated the sensor’s ability to identify concentration differences of water in isopropyl alcohol, but used a finer resolution—the solutions tested differed by a concentration of 2000 ppm = 0.2%.

##### Solutions Tested

Six liquid solutions of commercially available “99% isopropyl alcohol” (from the same lot) and deionized water, containing 0%, 0.2%, 0.4%, and 0.6% deionized water by volume in isopropyl alcohol were tested. In the following, these mixtures are referred to simply by their concentration of deionized water.

##### Preparation of Solutions

Four individual containers were used, each with a volume of 3940 mL. To make the 0% solution, the first container was filled with 99.9% isopropyl alcohol. To make the 0.2% solution, the second container was filled with isopropyl alcohol, after which 7.9 mL was removed and replaced with 7.9 mL deionized water. Other concentrations were prepared similarly, removing 15.8 mL and 23.6 mL, respectively, of isopropyl alcohol and replacing it with deionized water.

##### Identification of Blinded Samples

The team then followed the “Procedure for Blind Identification” described above using 1000.0 g of each of the prepared solutions.

#### 2.4.3. Experiment 3: Sodium Chloride in Deionized Water at 2000 ppm

The first two experiments involved identifying liquids mixed at different concentrations. Experiment 3, in turn, involved a saline solution and demonstrated the technology’s ability to quantify concentrations of sodium chloride in deionized water at a level of 2000 ppm.

##### Solutions Tested

Three solutions were prepared from deionized water and commercially available iodized table salt (sodium chloride) to contain 0%, 0.2%, and 0.4% sodium chloride (NaCl) by mass in deionized water. In the following, these mixtures are referred to simply by their concentration of sodium chloride.

##### Preparation of Solutions

Three containers were used, each with a mass of 3000.0 g deionized water. The 0.2% NaCl solution was prepared by removing 100 mL water, thoroughly mixing in 6 g NaCl and returning this to the container. The 0.4% solution was prepared similarly with 12 g NaCl.

##### Identification of Blinded Samples

The team then followed the “Procedure for Blind Identification” described above using 1000.0 g of each of the prepared solutions.

#### 2.4.4. Experiment 4: Commercial Bleach (NaOCl) in Deionized Water at 2000 ppm

Experiment 4 returned to a mixture of two solutions, and was designed to demonstrate the sensor’s ability to quantify concentrations of sodium hypochlorite (NaOCl) by volume in deionized water at a level of 2000 ppm.

##### Solutions Tested

Four solutions prepared from deionized water and sodium hypochlorite as found in commercially available bleach (Clorox^®^ Disinfecting Bleach (Clorox, Oakland, CA, USA), Sodium Hypochlorite Concentration 7.5%), containing 0%, 0.2%, 0.4%, and 0.6% NaOCl, respectively, were tested.

##### Preparation of Solutions

Four containers were used, each with a volume of 3940 mL. To make the 0% solution, the first container was filled with deionized water. To make the 0.2% solution, the second container was filled with deionized water, after which 105.2 mL was removed and replaced with bleach (105.2 mL bleach × 7.5% = 7.9 mL NaOCl). Other concentrations were prepared similarly, removing 210.4 mL and 315.6 mL, respectively, of deionized water and replacing it with bleach.

##### Identification of Blinded Samples

The “Procedure for Blind Identification” described above was then followed, using 1000.0 g of each of the prepared solutions.

#### 2.4.5. Experiment 5: Deionized Water in Isopropyl Alcohol at 10,000 ppm of a Fixed Volume

In each of the tests described above, the Know Labs team ensured consistency across scans of the analytes by using precisely 1000.0 g of each analyte for their scan. A potential drawback of this is that, since the density of each of the analytes is different, the volume was also different. This leads to a natural question—does the Bio-RFID technology really recognize molecular differences, or is it responding to only volumetric differences between the analytes? Experiment 5 was designed to demonstrate the sensor’s ability to quantify levels of water in isopropyl alcohol in the setting in which each solution had equal mass.

##### Solutions Tested

Experiment 5 consisted of the same analytes used in Experiment 1 (0–5% deionized water in isopropyl alcohol), in addition to 100% deionized water, using 1000 mL for each scan. To measure volume as precisely as possible, the mass of each analyte needed to acquire precisely 1000 mL was computed using the density of each solution, allowing for precision to within 0.1 g or ~0.1 mL. At a reference temperature of 25 °C, the density of isopropyl alcohol is 780.80 g/L, and the density of water is 997.07 g/L, giving rise to Table 1.

### 2.5. Data Collection

For each solution, data were collected on a continuous basis, using sweeps across the 1500 MHz–3000 MHz range at 0.2 MHz intervals, collecting values at 7501 frequencies. A full sweep of these frequencies took approximately 22.7 s, followed by a one-second pause between sweeps. A total of 10 sweeps were performed on each analyte.

## 3. Data Analysis

For each experiment, a training dataset was compiled by scanning each of the solutions with a total of ten sweeps, after which the mean reading from each frequency was recorded. When a blinded solution was later scanned in the test data, the system attempted to identify it by identifying its nearest neighbor from the training set. The distance to each member of the training set was calculated using the L^1^ norm (also called the Manhattan distance or the taxicab metric). Thus the distance between a scan of a solution in the test data and a given element of the training data is the sum of the absolute values of the differences of the sensor value, summed over all frequencies.

## 4. Results

For each of the five experiments, 100% of solutions in the test data were correctly identified. Details on each of the scans for each of the experiments follow.

### 4.1. Experiment 1: Deionized Water in Isopropyl Alcohol at 10,000 ppm

As stated above, each of the blinded solutions was correctly identified in the test data. Table 2 shows the distance between the 0% water solutions and all other solutions (in both the training and test data). The Bio-RFID signals across the frequency range are shown in Figure 1.

The distances between each of the blind solutions and each of the values in the training data are given in Table 2 and plotted in Figure 2. Note that here, and in the other figures below, the color used for each analyte in the bar graph matches the color used for that analyte in the corresponding spectral graph.

When calculating the distance between each scanned solution and the 0% water solution, a natural ‘matching’ was clear. For example, the distance to the 2% water solution was 1,917,048, while the distance to Blind-6 was 1,910,652. Since no other value was close, Blind-6 was identified as 2% water. Other identifications proceeded similarly. Further reflections on this table appear below in the Discussion section.

### 4.2. Experiment 2: Deionized Water in Isopropyl Alcohol at 2000 ppm

Once again, each of the blinded solutions was correctly identified in the test data. Table 3 shows the distance between the 0% water solutions and all other solutions (in both the training and test data). The Bio-RFID signals across the frequency range are shown in Figure 3.

The distances between each of the blind solutions and each of the values in the training data are given in Table 3 and plotted in Figure 4.

As in Experiment 1, when distances were calculated between each scanned solution and the 0% water solution, each blind solution had an obvious match in the training data. When the blinded solutions were listed by order of distance to the 0% water, the distance of each had a single value in the training data to which it was closest.

### 4.3. Experiment 3: Sodium Chloride in Deionized Water at 2000 ppm

Recall that in Experiment 3, various saline solutions were scanned by the sensor once to create the training data, after which, blinded solutions were scanned in an attempt to identify each. Each of the blinded solutions in the test data was correctly identified. The Bio-RFID signals across the frequency range are shown in Figure 5. The distance between pure water and each of the saline solutions and all other solutions (in both the training and test data) is given in Table 4 and plotted in Figure 6.

Unlike in Experiments 1 and 2, in which the Bio-RFID curves appear to change rather consistently with increasing concentrations of water in alcohol, the effect of adding NaCl was more unpredictable. Nevertheless, each of the blinded samples was correctly identified; as can be seen in Table 4, each of the blinded samples had an obvious match in the training data.

### 4.4. Experiment 4: Commercial Bleach (NaOCl) in Deionized Water at 2000 ppm

Experiment 4 returned to liquid solutions and involved identifying different concentrations of bleach (NaOCl) in water. Each NaOCl solution was scanned by the sensor once to create the training data, after which, blinded solutions were scanned in an attempt to identify each. Each of the blinded solutions in the test data was correctly identified. The Bio-RFID signals across the frequency range are shown in Figure 7. The distance between pure water and each of the other solutions (in both the training and test data) is given in Table 5 and plotted in Figure 8.

### 4.5. Experiment 5: Deionized Water in Isopropyl Alcohol at 10,000 ppm of a Fixed Volume

Experiment 5 revisited the question of the identification of solutions of varying concentrations of water in isopropyl alcohol. In this experiment, exactly one liter of each solution was scanned and thus the mass of each solution was different. This experiment was not blinded as experimenters had access to the mass of the analyte during the test. However, conclusions can still be drawn about the ability of the sensor to distinguish between these solutions. The Bio-RFID signals across the frequency range are shown in Figure 9. The distance between pure water and each of the other solutions (in both the training and test data) is given in Table 6 and plotted in Figure 10.

Note that the distance between each of the analytes and the reference (0% water) was comparable to that of the constant mass test (Experiment 1), suggesting that molecular differences between similar analytes can be seen regardless of whether the analytes have equal volumes or equal masses.

## 5. Discussion

The major finding of this study is that Bio-RFID has the capacity to be trained in the accurate detection of concentrations of substances in solution. This evolving technology, while still under development, already has the potential to impact a large number of industries. Some of the biological and non-biological implications are discussed below.

### 5.1. Implications for Resolution Abilities of the Sensor

It is interesting to speculate about the resolving power of the sensor, given the restrictions we placed on it in this study. In order to estimate this, let us take, as an example, the Blind-1 solution in Experiment 1. Table 2 shows that if the concentration of water in isopropyl alcohol differs by 1% (10,000 ppm), the distance between the two signals is slightly less than a million (864,963), and this relationship seems fairly linear. A distance of one million between two signals thus represents a concentration difference of about 10,000 ppm. Put another way, each 1 ppm difference results in a distance between signals of about 864,963/10,000 ≅ 86.5.

At the same time, the distance between a scan of the 0% solution in the training data and a second scan of the same solution in the test data was 33,157 (Table 2). Given an analyte, any other analyte whose distance was less than 33,157 would be indistinguishable from it. Taking a ratio between this and the value that differs by 10,000 ppm, we expect that the resolution limit of this system is about 33,157/864,963 ≅ 0.0383 × 10,000 ppm = 383 ppm. That is, a concentration difference of 383 ppm could be identified, even without repeating measurements or employing any machine learning techniques.

However, the resolving power may be greater than this. A similar calculation based on the results from Experiment 2 suggests a resolution limit of 19,475/156,608 × 2000 ppm ≅ 249 ppm of water in an alcohol solution. Likewise, we might expect a resolving power of 84,262/692,414 × 2000 ppm ≅ 243 ppm NaCl in water, and 96,840/2,026,160 × 4000 ≅ 191 ppm NaOCl bleach in water.

### 5.2. In Vitro Implications

The tests described in this report indicate the ability of the Bio-RFID technology to detect differences in liquid analytes in an in vitro setting. Such a technology could be used in a wide range of applications ranging from real-time quality assurance systems for industrial, chemical, pharmaceutical, and food production, to the identification of counterfeit products and the testing of municipal water supplies, to name a few.

There are many settings in which quantification of impurities in a liquid sample at a level of about 100 ppm would be useful. In some industrial processes, such as metalworking or chemical production, impurities in the water used in the process can negatively impact the quality of the final product. The presence of chloride ions in the water used for cooling and lubrication in metalworking can lead to corrosion and other problems. The limit for chloride ions and sulfates in metalworking water is typically around 150 ppm [14].

There are a significant number of industrial applications where this portable, inexpensive sensor could be employed, and can be investigated in future work.

### 5.3. In Vivo Implications

While the experiments described in this paper were performed in vitro, there may be implications of these results for the measurement of physiologically relevant molecules in vivo. One important molecule that could conceivably be quantified non-invasively in vivo is blood glucose. Based on the range of resolution limits calculated above, we might guess that the resolution of glucose is in this range of values—say, 250 ppm. (Note that this assumes no refinement of protocol nor that machine learning is used—see Limitations). Therefore, the techniques described in this paper may already be sufficient to detect blood glucose differences on the order to 250 ppm = 25 mg/dL. This is an encouraging initial assumption, and we expect that future work will continue to improve the resolution of this system.

## 6. Limitations of This Study

The methods used in this paper to quantify solutes in solution are arguably crude, and the resolution limit of this method can be taken as a lower bound for the sensor’s capabilities. The study is limited first by the fact that each solution in the training set was scanned only once—repeating the scan would provide a great deal more information to any model. The fact that a single scan was sufficient to identify 100% of analytes is therefore impressive. Additionally, a nearest neighbor model ignores almost all of the fine structure of the data, and reduces a comparison of two Bio-RFID curves to a single number. It is quite likely that more sophisticated techniques (Fast Fourier Transforms, Neural Networks, etc.) could quantify solutes much more precisely.

Another limitation is that each of the experiments described in this paper demonstrates only the ability to assess the concentration of a known analyte in a solution which contains only that specific analyte. The scope of this study was to isolate the effects of a single variable (concentration of solute), not to compare the Bio-RFID signatures of different analytes. However, a visual examination of Figure 3, Figure 5 and Figure 7 suggests that mixing 0.2% solute into a liquid results in very different Bio-RFID signatures.

## 7. Conclusions

Bio-RFID offers an accurate method for detection of substances in solution, providing the system is trained in detection of that particular molecule concentration. This approach may have important implications for non-invasive detection of physiologic analytes in vivo, for detection of impurities in solutions, and for quality assurance systems for industrial, chemical, pharmaceutical, and food production.

## Figures and Tables

**Figure 1 sensors-23-04817-f001:**
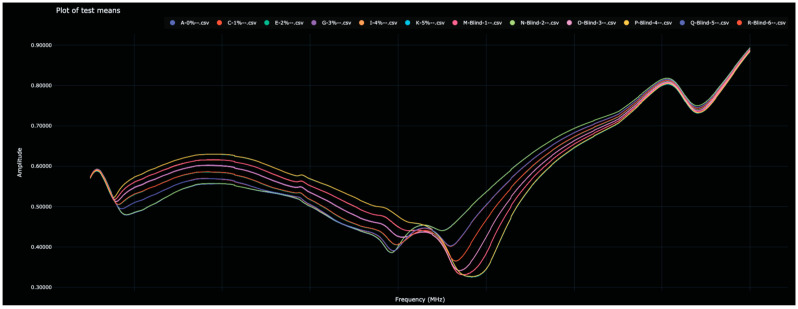
Display of the Bio-RFID signatures of analytes from Test 1, including the six training values and the six test values. Note, visually, that the twelve signatures appear as six curves due to the close match between training and test data.

**Figure 2 sensors-23-04817-f002:**
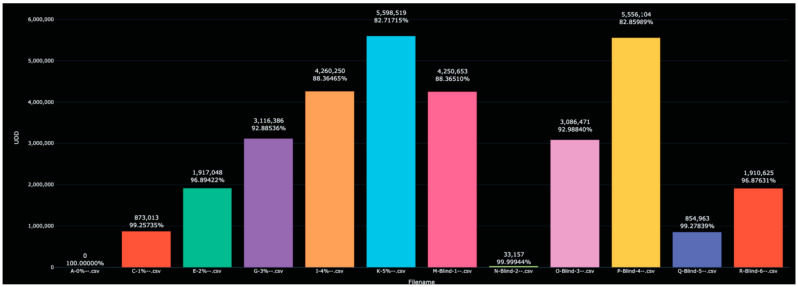
Display of the distance between the 0% water solution and all other scanned solutions in the test and training data, as measured in the L^1^ metric). Note that each blinded scan has an obvious match among the initial scans—the two recordings of the signature of the same analyte are almost identical and significantly different from the other scans.

**Figure 3 sensors-23-04817-f003:**
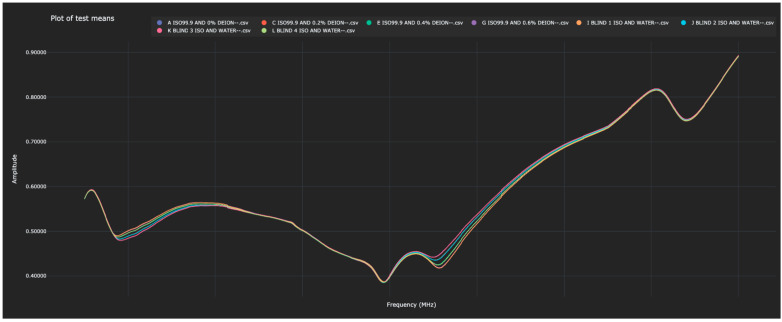
Display of the Bio-RFID signatures of analytes from Test 2, including the four training values and the four test values. Note that the eight signatures appear to the eye as four curves due to the close match between training and test data.

**Figure 4 sensors-23-04817-f004:**
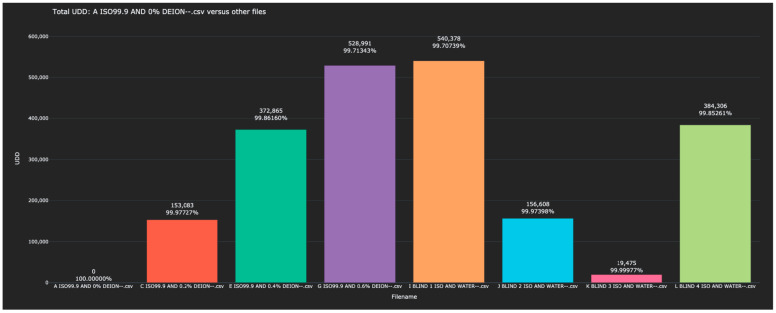
Display of the distance between the 0% water solution and all other scanned solutions in the test and training data, as measured in the L^1^ metric. Note that each blinded scan has an obvious match among the initial scans—the two recordings of the signature of the same analyte are almost identical and significantly different from the other scans.

**Figure 5 sensors-23-04817-f005:**
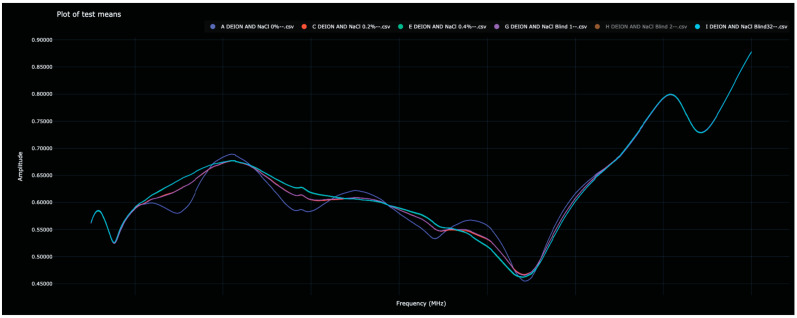
Bio-RFID signatures of 0%, 0.2%, and 0.4% NaCl in deionized water.

**Figure 6 sensors-23-04817-f006:**
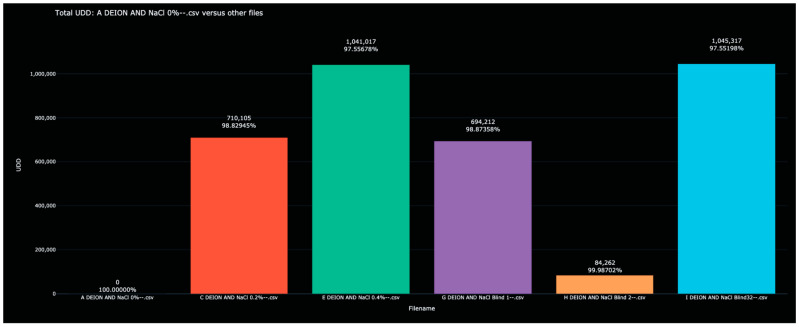
Display of the distance between the 0% saline solution and all other scanned saline solutions in the test and training data. Once again, each blinded scan has an obvious match among the initial scans.

**Figure 7 sensors-23-04817-f007:**
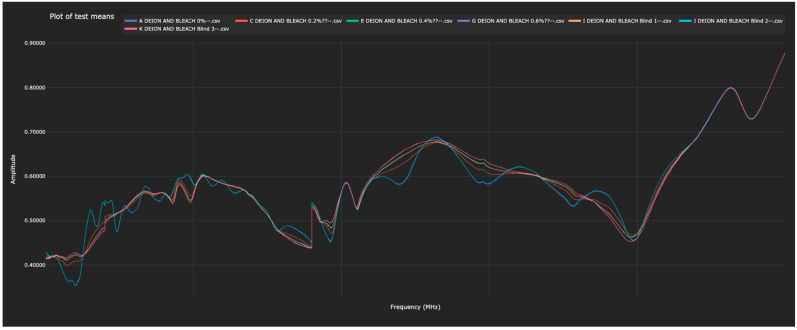
Bio-RFID signatures of bleach solutions corresponding to NaOCl concentrations of 0%, 0.2%, 0.4%, and 0.6% in deionized water.

**Figure 8 sensors-23-04817-f008:**
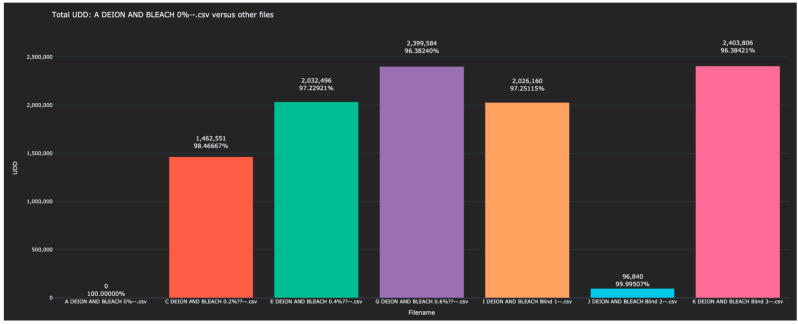
Display of the distance between the 0% water solution and all other scanned bleach solutions in the test and training data, as measured in the L1 metric).

**Figure 9 sensors-23-04817-f009:**
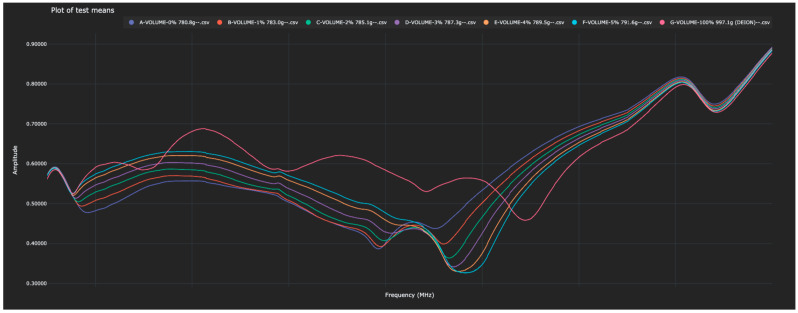
Bio-RFID signatures of 1 Liter water/isopropyl alcohol solutions with 0%, 1%, 2%, 3%, 4%, and 5% water by volume.

**Figure 10 sensors-23-04817-f010:**
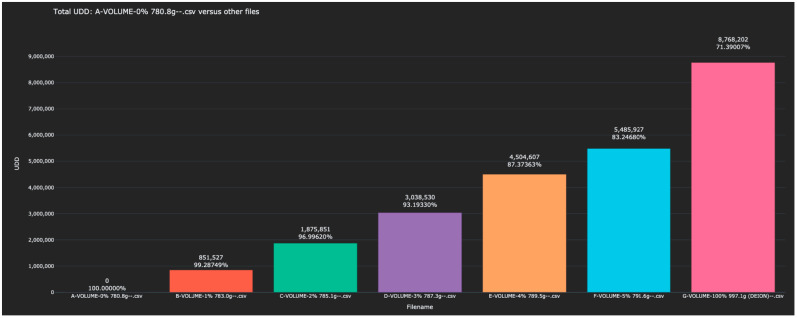
Display of the distance between the 0% water solution and all other scanned bleach solutions in the test and training data.

**Table 1 sensors-23-04817-t001:** Density of Water–Isopropyl Alcohol Solutions (25 °C reference).

% Water in Isopropyl Solution	Density g/L (Mass Needed for 1000 mL Volume)
0	780.8
1	783
2	785.1
3	787.3
4	789.5
5	791.6
100	997.1

Note that this volumetric method could not be used for Tests 1–4 that involved blinds, as the blind would be known to the Know Labs team by the mass required to generate the exact volume.

**Table 2 sensors-23-04817-t002:** Distance between 0% water and all scanned solutions.

Training Data	Distance	Test Data *	Distance
		Blind-2	33,157
1% Water	873,013	Blind-5	864,963
2% Water	1,917,048	Blind-6	1,910,652
3% Water	3,116,386	Blind-3	3,086,471
4% Water	4,260,250	Blind-1	4,250,653
5% Water	5,598,519	Blind-4	5,556,104

* Ordered by nearness.

**Table 3 sensors-23-04817-t003:** Distance between 0% water and all scanned solutions in Experiment 2.

Training Data	Distance	Test Data *	Distance
		Blind-3	19,475
0.2% Water	153,083	Blind-2	156,608
0.4% Water	372,865	Blind-4	384,306
0.6% Water	528,991	Blind-1	540,378

* Ordered by nearness.

**Table 4 sensors-23-04817-t004:** Distance between pure water and all scanned saline solutions in Experiment 3.

Training Data	Distance	Test Data *	Distance
		Blind-2	84,262
0.2% NaCl	710,105	Blind-1	694,212
0.4% NaCl	1,041,017	Blind-3	1,045,317

* Ordered by nearness.

**Table 5 sensors-23-04817-t005:** Distance between 0% water and all scanned bleach solutions in Experiment 4.

Training Data	Distance	Test Data *	Distance
		Blind-2	96,840
0.2% Bleach	1,462,551	Blind-3	no data
0.4% Bleach	2,032,496	Blind-4	2,026,160
0.6% Bleach	2,399,584	Blind-1	2,403,806

* Ordered by nearness.

**Table 6 sensors-23-04817-t006:** Distance between 0% water and all scanned alcohol solutions in Experiment 5.

Training Data	Distance
1% Water	851,527
2% Water	1,875,851
3% Water	3,038,530
4% Water	4,504,607
5% Water	5,485,927
100% Water	8,768,202

## Data Availability

Data are not available for reasons of intellectual property.

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
