# Peer review of "Detecting Unique Analyte-Specific Radio Frequency Spectral Responses in Liquid Solutions—Implications for Non-Invasive Physiologic Monitoring"

_sensors, 2023, doi:10.3390/s23104817_

Round 1
Reviewer 1 Report
The paper presents a very detailed in-vitro study to demonstrate, through a double-blind approach, the ability of the Bio-RFID sensor to detect analyte concentration in solutions.
The reported results definitely prove the sensor performance and the predicted sensitivity.
However, I have a few concerns regarding the conclusions and perspectives that the authors draw based on their findings.
1. Regarding "in vivo" implications, the authors mention the main application of non-invasively detecting blood glucose levels. From the experimental set-up described, it would be necessary to have blood samples available to perform the analysis, which would not be a non-invasive approach. I think the authors are envisaging an application scenario where the sensor is attached to the patient skin and somehow senses the blood below. But this is definitely not demonstrated by the present study. Moreover, the authors mention another study they performed (ref. [13], which is only submitted and, hence, unavailable to the reviewer) where they supposedly demonstrated in-vivo operation for blood glucose monitoring. Is this true? Then the present in-vitro study is not needed if they have already in-vivo evidence. Again, difficult to judge without reading ref. [13].
2. Concerning in-vitro industrial applications, I think that one of the main challenges would be to distinguish among different analytes. The present study only demonstrates the ability to assess the concentration of a KNOWN analyte in a solution which ONLY contains that specific analyte. This is a limitation which should be better discussed and highlighted.
Please also note that all the figures are unreadable (label font is really too small).
I think there is also a mistake in the caption of Fig. 3, which refers to Test 1 (I think it should read Test 2).
Reviewer 2 Report
Authors test an approach for detecting different analytes in vitro and using radio frequency identification technology. The technology is a promising one for non-invasive in vivo detection of biologically relevant analytes, such as glucose. The manuscript may be considered for publication, but requires major revision.
First of all, a more detailed description of the background should be provided describing the methods and results obtained in previous Bio-RFID experiments. Alternatively, the comparison of current results and methods and the previous ones can be provided in the Discussion.
Also, the introduction provides the detection of blood glucose levels as one of the goals for Bio-RFID technology, and maybe additional experiments with biologically relevant analytes can be provided in the manuscript, in addition to water, salt, and bleach.
Round 2
Reviewer 1 Report
The concerns raised during the previous review round have been correctly addressed.
I have no further comments.
Reviewer 2 Report
The authors have addressed all questions, and I think the manuscript can be accepted.